# Effects from ESG Scores on P&C Insurance Companies

**Silvia Bressan** 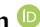

Department of Economics and Management, Free University of Bozen, 39100 Bozen, Italy; silvia.bressan@unibz.it

**Abstract:** Insurers act as institutional investors and underwriters of risk. Therefore, improving their environmental, social, and governance (ESG) performance is important for the transmission of ESG values to all economic sectors. We analyze ESG scores of worldwide Property and Casualty (P&C) insurers during 2013–2022 and show that more sustainable insurers have high operating leverage despite appearing to be financially stable from their combined ratios and z-scores. Additional results for the US subsample illustrate that stocks issued by sustainable insurers deliver positive excess returns. Overall, these findings suggest that there is a significant association between sustainable practices and the ability of insurers to execute business and create value. This is important for insurance managers, investors, and policy makers, as insurers play a prominent role in promoting economic growth and stability.

**Keywords:** insurance; ESG; sustainability

## 1. Introduction

The stable trend of equity capital flowing into environmental, social and governance (ESG) funds forces managers of insurance companies to deal with ESG strategies. The continued growth of "green" and sustainable funds means to insurers that they must actively monitor and promote their ESG ratings in order to retain full access to capital and manage the potential impacts on their stock prices. Thus, ESG practices would prevent firms from being excluded from ESG funds and indices (ESG is the term used most commonly in relation to investment products and underwriting. We use this terminology throughout the paper, and we call "sustainable insurers" firms that have high ESG ratings). To address sustainability risks, the United Nations Environment Programme Finance Initiative (UNEPFI) has produced a guide for the global insurance industry— the Principles for Sustainable Insurance (PSI), founded on the key principle that "insurers will incorporate ESG issues relevant to their business in their own decision making" (https://www.unepfi.org/insurance/insurance (accessed on 19 April 2023)).

Recently, the topic of sustainability has received much more attention from regulators and policy makers. Therefore, there is also vivid academic research on the effects of ESG on corporate finance. Scholars strive to measure the benefits from integrating ESG values in the management of corporations. One of the main issues is to establish whether high ESG profiles would enhance financial performance of companies. Theoretically, it has been proved that risk-averse investors have a preference for ESG, for example, by [1], and more recently by [2,3]. This implies that, in the cross-section, high-ESG firms are discounted at lower rates than low-ESG firms. In turn, this would mean that more sustainable companies enjoy cheaper financing costs while delivering high value to equity holders. Nonetheless, the empirical evidence is very heterogeneous, and the literature is inconclusive on the correlation between ESG scores and the performance of financial assets. Different articles prove that it may be positive, negative, or event nonexistent, depending on the samples under study and the methodologies implemented. Contributions in this field are numerous, and an exhaustive review can be found in [4] and, more recently, in [5].

In the existing literature, there is a lot of limited empirical evidence on the impact of ESG inside insurance companies, as the standard practice is to examine financial intermedi-

aries separately due to considerable differences in their business model and regulation. This gap of knowledge provides motivation to our goal of conducting an analysis focused on the insurance sector, i.e., a business playing a key role for the functioning of the entire economic system. Thus, studying financial strength of insurance companies and its relationship to ESG policies appears of paramount importance.

Some previous articles document the benefits from ESG in insurance. For example, in [6], the authors show that ESG scores enhance the stability of American insurers, while the authors of [7] illustrate that ESG rating upgrades generate abnormal stock returns inside European insurers. However, there is a lack of extensive evidence showing that ESG practices would effectively change the financial conditions of insurance companies. We now take up this challenge and analyze a sample of worldwide P&C insurers during 2013–2022. We find that ESG scores correlate positively with insurers' operating leverage while they are not significantly associated with financial leverage. At the same time, high ESG scores reveal low combined ratios and high z-scores. As we examine equity market valuations, we find that market-to-book ratios decrease in ESG scores. Finally, using data from the United States markets, we show that stocks issued by sustainable insurers earn highly positive excess returns.

We contribute to the literature on corporate sustainability providing new results from P&C insurance companies. Based on the evidence that ESG scores affect the risk–return trade-off in insurance [6,7], we now show that the ability of P&C insurers to achieve financial stability and create value is tightly linked to their sustainability practices.

Our findings are important for a few stakeholders. First, it is relevant for insurance managers to understand whether integrating ESG in decision-making would affect the fundamentals, providing an effective comparative advantage. Our results suggest that incorporating ESG would make insurers more resilient and highly valued. Arguably, this would decrease equity capital costs. The cost of equity capital is the rate of return that insurers have to pay for the equity they use. As this metric highly depends on the financial health of the company, a less sustainable insurer would be likely to face higher costs of financing. If the company does not pay the rate of return demanded, it may come under pressure from capital markets. The stock price may decrease quite sharply to the point that the company becomes a takeover target for competitors, or the management is removed [8].

Second, we deliver some key information for stock selection to portfolio managers. Based on our results, we suggest that the value of wealth portfolios would increase in the short term if they invested in the equity of high-ESG insurers, especially in the United States markets. Finally, regulators could employ the insights from our analysis to corroborate their efforts in addressing ESG values. In fact, policy makers encourage insurers worldwide to factor ESG in the underwriting process and investment decisions. As insurers contribute significantly to safeguard the economic system, it is an important task to collect evidence proving that sound ESG practices would make the insurance sector effectively more resilient.

The article is organized as follows. Section 2 presents the previous literature more closely related to our topic. Section 3 describes the data that we use for the analysis. Section 4 outlines the results. Section 5 shows additional results for the subsample of the United States insurers. Section 6 concludes.

## 2. Review of the Literature

The growing concerns of regulators and policy makers about ESG topics provide a strong incentive for academics to conduct research aimed at acquiring knowledge about the impact of sustainability on corporate finance decision making. Nonetheless, a large majority of the previous articles has addressed sustainability issues by analyzing data from non-financial firms or indices, while only few studies focused on the insurance sector. This, instead, is the focus of our article.

The author of [9] use a transparent framework to assess the integration of corporate social responsibility in the business of 153 international insurers in 2007. The author finds

a high level of variability across countries and firms, with social and ethical aspects of corporate social responsibility being better integrated than environmental aspects.

Only a few studies use the ESG scores of insurance companies to test their relationship with financial dimensions. For American insurers during 2006–2018, in [6], the authors illustrate that the increasing ESG scores diminish distress risk as measured by z-scores. The conclusion is that sustainability enhances the financial stability of insurers, while the data also reveal that this effect is stronger for the social and environmental pillars but not for the governance pillar. In [10], data are used from large United States insurers to implement a two-step method that first elaborates a measure for ESG awareness [11] and then relates such quantity to firm-specific characteristics. ESG awareness is found to be more significantly determined by the firm solvency, profitability, and size. In [12], sustainability is associated with the purchase of reinsurance. The author finds that high-ESG firms are more profitable and cede less risk to reinsurers. The author argues that sustainable insurers are less risky, and can save on reinsurance costs.

In the existing literature, the correlation between ESG scores and financial performance is found to be positive, negative, or even nonexistent (as in the survey in [4] about 2200 empirical papers that examine the association between ESG and corporate financial performance. The large majority of studies reports that ESG enhances performance in a stable way over time. More recently, in [13], findings were presented from a survey of 1141 papers and 27 meta-reviews published between 2015 and 2020, revealing that the financial performance of ESG investing is similar to conventional investing, with one third of studies showing superior performance). For the insurance industry, the evidence is very limited; therefore, it remains an open empirical issue to establish whether sustainable insurers deliver superior returns to investors. In study [7], the effect of ESG ratings on the stock price of European insurance companies is examined using an event study methodology. The results suggest that stock prices are highly responsive to ESG ratings, as the authors find that an upgrade in the ESG rating results in a significant stock price increase, while a rating downgrade leads to a decrease (The stock market reaction to responsible investments remains an open issue in the literature, as, for example, in [14], where it is shown that socially responsible investments of banks do not command positive market premiums. Another possible way through which ESG would affect stock prices could also be through incentives for earning management [15,16]. In [17], a mediation effect model is used to prove that market values of Chinese firms improve with ESG ratings through their increased operating capacity, while in [18], audit fees and controversies are among the channels that establish a link between the ESG ratings and the financial performance of European banks).

Following this stream of research, we analyze international P&C insurers and examine the relationship between ESG scores and measures of financial strength, in addition to stock market valuations. In this way, we integrate with new results the previous research addressing the hypothesis that sustainability creates value in the insurance industry.

## 3. Data and Variables

Using the Standard and Poor's Capital IQ platform, we assemble the data for our analysis following multiple steps. First, we select companies from all geographies (Africa, Asia-Pacific, Europe, Latin America and Caribbean, Middle East, United States and Canada) classified as "Property and Casualty (P&C) insurance". Inside this group, we keep only the operating and listed firms at the end of 2022. This means that companies that went bankrupt or were subject to mergers and acquisitions (M&As) before 2022 are excluded from our sample. In the category called "*ESG*", we obtain the *ESG* composite and partial (*E*, *S*, and *G*) scores. In the database, *ESG* scores are available with annual frequency starting from 2013 until 2022. As these firms are from worldwide regions, we download their accounting figures from the category called "S&Ps Global Universal Financials" (This information is provided based on the company filings, without adjusting for differences in the reporting rules across countries. We acknowledge that this is a potential caveat.

However, we would exclude the possibility that our results are primarily driven by different accounting standards. In the analysis, we use multiple (relative) measures for the financial health of the insurers; we also test our regressions clustering the standard errors by country. Although the coefficients have less statistical explanatory power, the signs do not change compared to the results we display in the paper. Therefore, this potential source of bias seems to be only marginally considerable). We finally assemble an unbalanced panel as reported in Table 1. Appendix A includes the complete list of the companies in the sample.

**Table 1.** Composition of the sample.

| Year | % of Sample | Geography | % of Sample |
|---|---|---|---|
| 2013 | 5.67 | Asia-Pacific | 31.19 |
| 2014 | 5.93 | Europe | 13.66 |
| 2015 | 6.70 | Latin America and the Caribbean | 1.55 |
| 2016 | 6.70 | Middle East | 0.52 |
| 2017 | 7.99 | The United States and Canada | 53.09 |
| 2018 | 7.99 | | |
| 2019 | 13.66 | | |
| 2020 | 14.43 | | |
| 2021 | 16.49 | | |
| 2022 | 14.43 | | |
| Total N observations | 390 | | |

S&P Capital IQ computes the ESG score as a discrete number in the range of 0–100 reflecting the performance of the company on key environmental, social, and governance issues according to an industry-specific assessment methodology and aggregation schemes. Higher values of ESG scores indicate stronger performance in sustainability practices (The S&P ESG ratings are based on the Corporate Sustainability Assessment (CSA), in addition to the information provided directly to S&P and certified by analysts, as well as on public domain information. More information on the methodology employed by S&P Capital IQ to compute ESG ratings can be found at https://www.spglobal.com/esg/csa/methodology/ (accessed on 19 April 2023)). In our analysis, we denote the composite score *ESG* while we also exploit the granularity of our database to decompose the single pillars, calling *E*, *S*, and *G* the three separate components. In the following regressions, *ESG*, *E*, *S*, and *G* are our independent variables, besides a few additional controls that we outline below in the text. Descriptive statistics in Table 2 show that the governance dimension has the best rating in the sample, with median *G* equal to 35. Instead, median *E* and *S* are smaller and respectively equal to 16 and 15 (Sometimes, in empirical research the ESG ratings are taken as logarithms, like, for example, in [19]. Following the majority of the studies in this field, we opt for including in our regressions the levels of the ESG scores. However, we check that we obtain higher coefficients as we use natural logarithms, but the quality of the outcomes does not change).

We test the relationship between ESG scores and variables that approximate important concepts in the financial analysis of P&C insurers. First, we briefly outline the P&C insurance business and then describe our variables. Table A1 in Appendix A summarizes their definition.

P&C insurance is one type of non-life insurance that protects against property losses and/or against legal liability that may result from injury or damage to the property of others. Some examples of P&C insurance products include motor insurance, homeowner (or renter) insurance and flood insurance, or natural catastrophe insurance in general. The success of a P&C insurer depends quite substantially on its ability to correctly segment and price the risks it underwrites, pool these risks, and optimize operating costs. Besides the premiums earned, an additional revenue line to P&C insurers stems from the financial returns obtained through their investments, which has an impact on their risk-taking and on premiums. In order to assess the risk-taking of our insurers, we approximate their

leverage with the following two measures, which take into consideration the source of the risk undertaken. We define *OL* the operating (or underwriting) leverage ratio, namely net premiums written divided by the surplus of policyholders. This number is called "surplus ratio" by practitioners, and in the literature, *OL* is employed as a fundamental measure of leverage inside insurance companies [20]. At the *OP* denominator, the policyholder surplus is the amount by which a company's assets exceed its liabilities, and it is the asset cushion which an insurance company maintains to protect itself, its policyholders and its shareholders against losses and adverse conditions. Thus, *OL* measures the insurer's exposure to pricing errors in its current book of business. For example, in [21], *OL* is used to approximate insurers' risk-taking behaviors and investigate joint dynamics with the loss ratio in order to identify structural changes in the underwriting cycles of property and liability (P&L) insurers. In the United States, the premium to surplus ratio is among the tests used in the Insurance Regulatory Information System (IRIS) to predict insurers' financial strength in order to prevent insolvency. A premium to surplus ratio of more than three indicates that the insurer operates with high underwriting risk and would not be considered solvent (https://content.naic.org/sites/default/files/publication-uir-zb-iris-ratios-manual.pdf (accessed on 19 April 2023)). On average, our firms seem to be financially strong, as the median *OL* inside Table 2 equals to 1.07. For robustness, we also compute *PRE* as the ratio of net premiums written to stockholders' equity. The quantity is inversely related to financial strength, as high values of *PRE* reveal that stockholders are highly exposed to operating risks.

**Table 2.** Descriptive statistics. See Appendix A, Table A1 for the definitions of all variables included in the models.

|  | **Mean** | **Median** | **Min** | **Max** | **St. Dev.** |
|---|---|---|---|---|---|
| Variables |  |  |  |  |  |
| *ESG* | 31.54 | 22.5 | 0 | 85 | 23.19 |
| *E* | 27.56 | 16 | 0 | 97 | 29.97 |
| *S* | 25.1 | 15 | 0 | 86 | 25.66 |
| *G* | 38.71 | 35 | 1 | 87 | 20.89 |
| *OL* | 1.12 | 1.07 | 0.04 | 4.45 | 0.74 |
| *PRE* | 1.19 | 1.10 | −0.04 | 4.79 | 0.66 |
| *FL* | 0.004 | 0.003 | 0 | 0.080 | 0.01 |
| *CR* | 0.95 | 0.94 | 0.75 | 1.37 | 0.08 |
| *ZSCORE* | 0.93 | 0.59 | 0.04 | 10.82 | 1.22 |
| *MTB* | 1.93 | 1.33 | 0.037 | 10.81 | 1.77 |
| *SIZE* | 16.95 | 17.13 | 10.29 | 21.04 | 1.60 |

The second dimension of leverage is the financing leverage, which we approximate by dividing the company's debt by the capital raised by issuing stocks and equity. Therefore, the financial leverage ratio *FL* measures the degree to which the firm is vulnerable to investment-related risks like interest rate and credit risks.

We calculate the combined ratio (also called "insurance ratio", or "Kenney ratio"), which is a measure widely employed in the industry to evaluate the profitability and financial health of an insurance company. The quantity *CR* is the sum of incurred losses, loss adjustment expenses plus other underwriting expenses divided by earned premiums. Thus, *CR* represents the amount of losses and underwriting expenses paid out per dollar of premium. A combined ratio below 100% indicates that the insurer is making underwriting profit, while a ratio above 100% indicates an underwriting loss. In the literature, the combined ratio is often used to approximate insolvency rates, proving that patterns in insurers' combined ratios also predict the underwriting cycles [22–24].

We employ the z-score as an indicator for the firm stability. The quantity *ZSCORE* is the sum of equity to total assets and return on average assets scaled by the three-year standard deviation of the average return on assets [6]. The z-score is computed

by researchers and practitioners to gauge the financial strength of insurance firms, as increasing values of $ZSCORE$ reveal lowering distress probability [6,25–27].

The market-to-book ratio $MTB$ is the ratio of market value of equity to book value of equity. This is an indicator that practitioners widely use in order to assess the capacity of a firm to generate value for shareholders. In fact, a ratio of one indicates that shareholders can only expect a return of book value. A ratio below one is typical for so-called "value companies", meaning that investors could buy the firm stock for a low price relative to the value of its assets. A ratio above one instead indicates that the firm is overvalued. In the literature, the market-to-book ratio is frequently used to capture the firm's growth opportunities [28]. Firms with a high $MTB$ (so-called "growth companies") entail high default risk, as they are usually more volatile with respect to the existing future growth opportunities. For example, in [29], the market-to-book ratio is computed on a sample of insurers to investigate the link between default risk and executive compensation.

Table 3 reports pairwise correlation coefficients among our variables. In the next section, we test the hypothesis that ESG scores have an impact on the insurer's financial strength by implementing regressions summarized with the following Equation (1), where the subscripts $j$ and $t$ denote, respectively, the company and the year:

$$Financial\ strength_{s,j,t} = \alpha_0 + \alpha_1 ESG_{,j,t-1} + \alpha_2 SIZE_{,j,t-1} + \tau_t + \omega_{j,t}. \tag{1}$$

Subscript $s$ indicates the variable capturing insurer $j$'s financial strength in year $t$, i.e., $OL$, $PRE$, $FL$, $CR$, $ZSCORE$, and $MTB$. $SIZE$ is our control variable for the corporate size that we measure by taking the natural logarithm of total assets. $\tau_t$ denotes time fixed effects, while $\omega_{j,t}$ is the error term.

**Table 3.** Correlation. See Appendix A, Table A1 for the definitions of all variables included in the models. *** $p < 0.01$, ** $p < 0.05$, * $p < 0.1$.

| | ESG | E | S | G | OL | PRE | SA | FL | CR | ZSCORE | MTB | SIZE |
|---|---|---|---|---|---|---|---|---|---|---|---|---|
| ESG | 1.000 | | | | | | | | | | | |
| E | 0.964 *** | 1.000 | | | | | | | | | | |
| | (0.000) | | | | | | | | | | | |
| S | 0.979 *** | 0.947 *** | 1.000 | | | | | | | | | |
| | (0.000) | (0.000) | | | | | | | | | | |
| G | 0.951 *** | 0.870 *** | 0.887 *** | 1.000 | | | | | | | | |
| | (0.000) | (0.000) | (0.000) | | | | | | | | | |
| OL | 0.235 *** | 0.236 *** | 0.294 *** | 0.157 ** | 1.000 | | | | | | | |
| | (0.000) | (0.000) | (0.000) | (0.006) | | | | | | | | |
| PRE | 0.162 ** | 0.149 ** | 0.213 *** | 0.104 | 0.974 *** | 1.000 | | | | | | |
| | (0.005) | (0.010) | (0.000) | (0.074) | (0.000) | | | | | | | |
| FL | −0.040 | −0.044 | −0.040 | −0.035 | −0.182 ** | −0.114 | −0.268 *** | 1.000 | | | | |
| | (0.469) | (0.424) | (0.464) | (0.523) | (0.002) | (0.053) | (0.000) | | | | | |
| CR | −0.116 * | −0.083 | −0.092 | −0.164 ** | 0.282 *** | 0.167 ** | −0.262 *** | 0.018 | 1.000 | | | |
| | (0.038) | (0.138) | (0.102) | (0.003) | (0.000) | (0.005) | (0.000) | (0.756) | | | | |
| ZSCORE | 0.108 * | 0.128 * | 0.097 | 0.120 * | −0.128 * | −0.051 | 0.034 | −0.042 | −0.086 | 1.000 | | |
| | (0.047) | (0.019) | (0.076) | (0.028) | (0.026) | (0.387) | (0.534) | (0.449) | (0.132) | | | |
| MTB | −0.223 *** | −0.260 *** | −0.207 *** | −0.203 *** | 0.003 | 0.160** | 0.011 | −0.090 | −0.291 *** | −0.069 | 1.000 | |
| | (0.000) | (0.000) | (0.000) | (0.000) | (0.961) | (0.007) | (0.847) | (0.109) | (0.000) | (0.217) | | |
| SIZE | 0.259 *** | 0.264 *** | 0.231 *** | 0.292 *** | 0.114 * | 0.113 | −0.360 *** | 0.077 | −0.031 | 0.110 * | −0.214 *** | 1.000 |
| | (0.000) | (0.000) | (0.000) | (0.000) | (0.045) | (0.052) | (0.000) | (0.158) | (0.586) | (0.043) | (0.000) | |

## 4. Results

Following Equation (1), columns (1)–(4) of Table 4 report the outcomes from pooled OLS regressions of $OL$ on the one period lagged $ESG$, and the partial scores $E$, $S$, and $G$. All regressions control for $SIZE$, time fixed effects, and a constant term. T-statistics are reported

in parentheses and the standard errors are robust. We observe that the coefficients are all positive, although the social and environmental dimensions are statistically significant, while the governance pillar is not significant. In columns (5)–(8), the regressions for *PRE* exhibit a similar pattern. These findings suggest that high-ESG firms are more exposed to underwriting risk, as they write a huge amount of policies without proportionally raising the capital to cover potential losses. In columns (9)–(12), the coefficients in the regressions estimated for financial leverage *FL* are negative but never significant. Thus, sustainability does not have a considerable impact on the capital structure of insurers.

**Table 4.** Results of OLS models with dependent variables *OL*, *PRE*, and *FL* in $t + 1$. See Appendix A, Table A1 for the definitions of all variables included in the models. Controls include *SIZE*, time dummies, and a constant term. T-statistics are reported in parentheses and standard errors are robust. *** $p < 0.01$, ** $p < 0.05$, * $p < 0.1$.

| Regressors | (1) OL | (2) OL | (3) OL | (4) OL | (5) PRE | (6) PRE | (7) PRE | (8) PRE | (9) FL | (10) FL | (11) FL | (12) FL |
|---|---|---|---|---|---|---|---|---|---|---|---|---|
| ESG | 0.0076 *** (2.958) | | | | 0.0043 ** (2.060) | | | | −0.0014 (−0.737) | | | |
| E | | 0.0058 *** (3.060) | | | | 0.0028 * (1.805) | | | | −0.0012 (−0.847) | | |
| S | | | 0.0092 *** (3.997) | | | | 0.0050 *** (2.787) | | | | −0.0012 (−0.727) | |
| G | | | | 0.0040 (1.280) | | | | 0.0035 (1.348) | | | | −0.0013 (−0.530) |
| Controls | Yes | Yes | Yes | Yes | Yes | Yes | Yes | Yes | Yes | Yes | Yes | Yes |
| N. of obs. | 309 | 309 | 309 | 309 | 309 | 309 | 309 | 309 | 309 | 309 | 309 | 309 |
| R−squared | 0.068 | 0.070 | 0.094 | 0.041 | 0.046 | 0.042 | 0.060 | 0.188 | 0.019 | 0.019 | 0.019 | 0.018 |

In columns (1)–(4) of Table 5, we find that the combined ratio *CR* decreases in ESG scores, with *E* and *G* also being more significant. The combined ratio is inversely related to the company's financial strength; therefore, high ESG firms seem to have solid financial fundamentals. The estimates for the z-score reveal consistent outcomes, as in columns (5)–(8) the coefficients on *ZSCORE* are positive, and again significant for *E* and *G*. Finally, firms with a stronger ESG performance have lower market-to-book ratios (see columns (9)–(12)).

**Table 5.** Results of OLS models with dependent variables *CR*, *ZSCORE*, and *MTB* in $t + 1$. See Appendix A, Table A1 for the definitions of all variables included in the models. Controls include *SIZE*, time dummies, and a constant term. T-statistics are reported in parentheses and standard errors are robust. *** $p < 0.01$, ** $p < 0.05$, * $p < 0.1$.

| Regressors | (1) CR | (2) CR | (3) CR | (4) CR | (5) ZSCORE | (6) ZSCORE | (7) ZSCORE | (8) ZSCORE | (9) MTB | (10) MTB | (11) MTB | (12) MTB |
|---|---|---|---|---|---|---|---|---|---|---|---|---|
| ESG | −0.0005 ** (−2.037) | | | | 0.0040 (1.587) | | | | −0.0156 *** (−3.147) | | | |
| E | | −0.0003 * (−1.686) | | | | 0.0037 * (1.916) | | | | −0.0139 *** (−3.747) | | |
| S | | | −0.0003 (−1.562) | | | | 0.0033 (1.500) | | | | −0.0126 *** (−2.902) | |
| G | | | | −0.0007 *** (−2.657) | | | | 0.0054 * (1.668) | | | | −0.0178 *** (−2.811) |
| Controls | Yes | Yes | Yes | Yes | Yes | Yes | Yes | Yes | Yes | Yes | Yes | Yes |
| Num. of obs. | 309 | 309 | 309 | 309 | 309 | 309 | 309 | 309 | 309 | 309 | 309 | 309 |
| R-squared | 0.063 | 0.058 | 0.057 | 0.074 | 0.073 | 0.077 | 0.072 | 0.074 | 0.072 | 0.085 | 0.066 | 0.065 |

To summarize our findings, we first note that the operating leverage increases with ESG scores. That is, the underwriting capacity of firms decreases as they exhibit a better ESG performance. However, it seems that sustainability does not have a substantial effect on the capital structure. Our interpretation is that the cost of debt does not change considerably while the ESG profile improves, in line with the results in [30] (In previous research, it is not well established whether debt issuances would grow with ESG practices. For example, in [31,32] it is found that ESG disclosure and ESG scores both have a negative effect on the cost of debt financing. In contrast, in [33], it is argued that the cost of debt increases when ESG disclosures interact with growth opportunities to reveal prospective risk. In [34], it is shown that the corporate sustainability performance has an impact on capital structure dynamics by increasing the speed at which firms adjust their leverage ratios to the target levels. To investigate whether ESG scores influence dynamics in the financial leverage of our firms, we tested the regressions of annual changes in *FL* on lagged ESG scores. In line with the outcomes of Table 4, the coefficients on *ESG* and on the three pillars are never statistically significant. These results are available upon request).

At the same time, however, we find that high-ESG insurers are more resilient, as increasing ESG scores are associated with low combined ratios and high z-scores (Note that all of our regressions control for time fixed effects could capture potential time variation in the estimated relationships. We also tested that the outcomes in Tables 4 and 5 do not change substantially in quality as we make *ESG* interact with dummies that identify (i) the COVID-19 pandemic in year 2020 or (ii) the European Union Directive on Non-Financial Reporting (2014/95/EU) in year 2014 that requires companies to include non-financial statements (including information on ESG policies) in their annual reports or in a separate filing from 2018 onward. As this directive applies to public interest companies with more than 500 employers in the European Union, it could have potential effects on our sample. Nonetheless, our estimates do not change significantly in 2018, even as we consider separately the subsample of European insurers. These results are available upon request). Our findings are in line with previous evidence in [6] related to American insurers during 2006–2018. In fact, the authors document a positive effect from ESG scores on z-scores, while they also discover that the relationship is statistically significant for the environmental and the social pillars. In the case of our worldwide insures, we find instead that the environmental and the governance dimensions are the most relevant in explaining financial strength.

Overall, the outcomes reveal that the financial strength of insurers interacts with sustainability. Although insurers with sound ESG profiles seem to face higher risks related to their operating business, the firms do not appear to face increasing investment risks. Our indicators for financial strength suggest instead that high-ESG insurers are financially resilient. Moreover, as we use the market-to-book ratio as a metric for equity valuation, we conclude that increasing sustainability enhances the "value" feature of insurer equity rather than the "growth" one. In order to examine more deeply the stock market performance of our companies, in the next section, we analyze stock market returns for the subsample of the United States insurers.

## 5. Stock Returns of US Insurers

For the subsample of the United States companies, we obtain from Compustat the market data with monthly frequency. These firms represent approximately 53% of our sample (see Table 1), and the firms are traded on the NYSE or the NASDAQ. In Table 6, we conduct pooled regressions of stock returns (*r*) on *ESG* scores plus the factors in [35] (i.e., market excess return *MKTRF*, small-minus-big *SMB*, and high-minus-low *HML*), the momentum factor (*MOM*) as in [36], and time dummies. We find positive coefficients on *ESG* as well on the three pillars, yet only *G* is significant. (Factors *MKTRF*, *SMB*, *HML* and *MOM* are taken from the Kenneth R. French's data library at https://mba.tuck.dartmouth.edu/pages/faculty/ken.french/data_library.html (accessed on 30 April 2023). We

checked that the quality of the results does not change when all regressors are lagged by one period. These results are available upon request).

**Table 6.** Results of OLS models with dependent variable $r$ in $t+1$. See Appendix A, Table A1 for the definitions of all variables included in the models. Time dummies and a constant term are included but not reported. T-statistics are reported in parentheses and standard errors are robust. *** $p < 0.01$, ** $p < 0.05$, * $p < 0.1$.

| Regressors | (1) $r$ | (2) $r$ | (3) $r$ | (4) $r$ |
|---|---|---|---|---|
| ESG | 0.0313 (1.568) | | | |
| E | | 0.0219 (1.562) | | |
| S | | | 0.0208 (1.007) | |
| G | | | | 0.0426 ** (2.078) |
| MKTRF | 4.0839 *** (4.486) | 0.6076 *** (13.194) | 4.0968 *** (4.499) | 4.0529 *** (4.453) |
| SMB | −0.3826 (−0.353) | −0.0574 (−0.686) | −0.3280 (−0.302) | −0.5145 (−0.473) |
| HML | −7.0256 *** (−3.663) | 0.3226 *** (6.160) | −7.0591 *** (−3.680) | −6.9445 *** (−3.621) |
| MOM | 7.2387 *** (3.832) | 0.0884 (1.355) | 7.2373 *** (3.830) | 7.2421 *** (3.836) |
| Num. of obs. | 2124 | 2124 | 2124 | 2124 |
| R-squared | 0.208 | 0.108 | 0.207 | 0.208 |

However, in the asset pricing literature, the standard approach is to analyze portfolio returns. Therefore, we rank our stocks into quintiles of *ESG* and form value-weighted portfolios, i.e., the portfolio holdings are proportional to the stock market capitalization computed as average stock price times average number of outstanding shares. Portfolios of insurers in the first quintile are the least sustainable assets, while the fifth quintile corresponds to the most sustainable portfolios. In Table 7, we perform regressions in each quintile to compare the portfolio value-weighted return ($r_p$) to the performance of the benchmark market model.

**Table 7.** Results of OLS models for the returns of *ESG* quintile portfolios. See Appendix A, Table A1 for the definitions of all variables included in the models. T-statistics are reported in parentheses and standard errors are robust. The coefficients on $\alpha$ are in percentages. *** $p < 0.01$, ** $p < 0.05$, * $p < 0.1$.

| | Portfolio Quintile | | | | | |
|---|---|---|---|---|---|---|
| | (1) $r_p$ | (2) $r_p$ | (3) $r_p$ | (4) $r_p$ | (5) $r_p$ | (5)–(1) $r_p$ |
| MKTRF | 0.751 *** (9.821) | 0.842 *** (8.854) | 0.551 ** (3.287) | 0.739 *** (6.465) | 0.757 *** (7.384) | 0.00666 (0.069) |
| HML | 0.480 *** (5.784) | 0.261 ** (3.051) | 0.292 (1.613) | 0.496 *** (3.419) | 0.408 *** (3.495) | −0.0726 (−0.855) |
| SMB | 0.0353 (0.225) | −0.127 (−0.991) | 0.172 (0.984) | −0.0617 (−0.349) | −0.383 ** (−2.645) | −0.418 *** (−3.585) |
| MOM | 0.181 (1.757) | 0.524 *** (5.039) | 0.0115 (0.062) | 0.295 (1.971) | 0.289 * (2.294) | 0.107 (0.898) |
| $\alpha$ | −0.113 (−0.344) | 0.176 (0.446) | 0.922 (1.874) | 0.286 (0.698) | 0.885 * (2.351) | 0.998 ** (2.640) |
| N. of obs. | 120 | 84 | 96 | 120 | 120 | 120 |

The coefficient on the constant term $\alpha$ measures the excess (or abnormal) return above the benchmark return. Our regressions estimate a significant $\alpha$ in the fifth quintile, i.e., the most sustainable portfolios outperform the market. In column (6) of Table 7, we consider a long/short position that is as follows: long USD 1 in a portfolio of the high-ESG firms and short USD 1 in a portfolio of the low-ESG firms. This strategy delivers a positive excess return measured by $\alpha$ close to 1%. As we conduct separate portfolio tests (see Tables 8–10), we find that the governance pillar drives the results that we observed for the composite rating. In fact, a long/short strategy on stocks with high/low governance delivers a significant $\alpha$ close to 1.17% (see Table 10).

**Table 8.** Results of OLS models for the returns of *E* quintile portfolios. See Appendix A, Table A1 for the definitions of all variables included in the models. T-statistics are reported in parentheses and standard errors are robust. The coefficients on $\alpha$ are in percentages. *** $p < 0.01$, ** $p < 0.05$, * $p < 0.1$.

| | Portfolio Quintile | | | | | |
| --- | --- | --- | --- | --- | --- | --- |
| | **(1)** | **(2)** | **(3)** | **(4)** | **(5)** | **(5)–(1)** |
| | $r_p$ | $r_p$ | $r_p$ | $r_p$ | $r_p$ | $r_p$ |
| *MKTRF* | 0.765 *** | 1.254 *** | 0.647 ** | 0.644 *** | 0.740 *** | −0.0252 |
| | (8.970) | (5.674) | (3.127) | (5.444) | (6.668) | (−0.288) |
| *HML* | 0.318 *** | 0.397 | 0.210 | 0.224 | 0.548 *** | 0.230 * |
| | (3.712) | (1.733) | (0.961) | (1.650) | (4.954) | (2.438) |
| *SMB* | −0.0585 | −0.392 * | 0.0262 | −0.0563 | −0.194 | −0.135 |
| | (−0.531) | (−2.293) | (0.085) | (−0.294) | (−1.283) | (−1.119) |
| *MOM* | 0.212 * | 0.0936 | 0.237 | 0.412 * | 0.196 | −0.0154 |
| | (2.431) | (0.334) | (0.997) | (2.240) | (1.573) | (−0.155) |
| $\alpha$ | 0.183 | −0.593 | 0.717 | 1.18 * | 0.458 | 0.275 |
| | (0.681) | (−1.190) | (1.050) | (2.510) | (1.184) | (0.797) |
| Num. of obs. | 120 | 36 | 60 | 96 | 120 | 120 |

**Table 9.** Results of OLS models for the returns of *S* quintile portfolios. See Appendix A, Table A1 for the definitions of all variables included in the models. T-statistics are reported in parentheses and standard errors are robust. The coefficients on $\alpha$ are in percentages. *** $p < 0.01$, ** $p < 0.05$, * $p < 0.1$.

| | Portfolio Quintile | | | | | |
| --- | --- | --- | --- | --- | --- | --- |
| | **(1)** | **(2)** | **(3)** | **(4)** | **(5)** | **(5)–(1)** |
| | $r_p$ | $r_p$ | $r_p$ | $r_p$ | $r_p$ | $r_p$ |
| *MKTRF* | 0.731 *** | 0.911 *** | 0.725 *** | 0.746 *** | 0.733 *** | 0.00257 |
| | (6.640) | (11.031) | (6.485) | (6.747) | (5.883) | (0.029) |
| *HML* | 0.289 * | 0.495 *** | 0.394 ** | 0.339 * | 0.503 *** | 0.214 |
| | (2.271) | (5.618) | (2.994) | (2.455) | (3.930) | (1.550) |
| *SMB* | 0.0933 | −0.146 | −0.0477 | −0.228 | −0.0221 | −0.115 |
| | (0.662) | (−0.921) | (−0.306) | (−1.173) | (−0.144) | (−0.713) |
| *MOM* | 0.315 ** | 0.162 | 0.244 | 0.383 * | 0.309 * | −0.00534 |
| | (2.789) | (1.200) | (1.594) | (2.212) | (2.289) | (−0.044) |
| $\alpha$ | 0.052 | 0.038 | 0.837 | 0.380 | 0.773 | 0.721 |
| | (0.142) | (0.091) | (1.976) | (0.954) | (1.852) | (1.610) |
| Num. of obs. | 120 | 96 | 108 | 108 | 120 | 120 |

**Table 10.** Results of OLS models for the returns of *G* quintile portfolios. See Appendix A, Table A1 for the definitions of all variables included in the models. T-statistics are reported in parentheses and standard errors are robust. The coefficients on $\alpha$ are in percentages. *** $p < 0.01$, ** $p < 0.05$, * $p < 0.1$.

| | Portfolio Quintile | | | | | |
|---|---|---|---|---|---|---|
| | **(1)** | **(2)** | **(3)** | **(4)** | **(5)** | **(5)–(1)** |
| | $r_p$ | $r_p$ | $r_p$ | $r_p$ | $r_p$ | $r_p$ |
| *MKTRF* | 0.797 *** | 0.903 *** | 0.533 ** | 0.691 *** | 0.736 *** | −0.0616 |
| | (7.431) | (9.571) | (3.240) | (6.233) | (6.636) | (−0.482) |
| *HML* | 0.376 *** | 0.534 *** | 0.246 | 0.466 *** | 0.338 ** | −0.0387 |
| | (4.116) | (4.992) | (1.220) | (4.306) | (2.677) | (−0.323) |
| *SMB* | −0.124 | −0.0863 | −0.0383 | −0.168 | −0.168 | −0.0437 |
| | (−0.962) | (−0.618) | (−0.233) | (−1.082) | (−1.026) | (−0.329) |
| *MOM* | 0.283 ** | 0.396 ** | 0.170 | 0.157 | 0.407 ** | 0.124 |
| | (2.643) | (3.168) | (1.020) | (1.189) | (2.832) | (0.891) |
| $\alpha$ | −0.163 | 0.385 | 0.768 | 0.553 | 1.000 * | 1.170 ** |
| | (−0.472) | (1.051) | (1.577) | (1.479) | (2.550) | (2.701) |
| Num. of obs. | 120 | 108 | 108 | 120 | 120 | 120 |

The results for the United States subsample in Table 7, together with the results in Table 5, are consistent with those of previous articles on the "value premium" in stock returns. Documented among the first in [37], the value premium is a well-established finding in the finance literature, and it asserts that companies with low market-to-book ratios (i.e., value companies) yield higher returns than companies with higher ratios (i.e., growth companies). For the subsample of the United States insurers, we preliminary checked that *MTB* decreases significantly with ESG scores, as it was for the whole sample in Table 5. (These results are available upon request. However, the correlation between ESG scores and the market-to-book ratio remains an empirical issue, and, for example, in [38], ESG rating quintiles are compared within the MSCI World Index over a 10-year period in 2007–2017, showing that the highest quintile also has the highest market-to-book ratio). Therefore, based on these outcomes, we put forward the argument that ESG scores plausibly contribute to determine value premiums inside the United States insurers.

## 6. Conclusions

The growing ESG concerns across all economic sectors pose important challenges to insurance companies, and it is important to determine whether ESG scores are informative for insurer financial strength and market performance. We analyze worldwide P&C insurers and show that ESG scores affect, in a significant way, a few indicators (leverage ratios, combined ratio, z-score, market-to-book ratio) that are widely employed by practitioners and asset managers to assess investment trade-offs. High-ESG insurers seem to be financially healthy, although exposed to an increasing underwriting risk. Furthermore, they earn abnormal returns in the stock market. These findings deliver important insights to insurer stakeholders, i.e., policy makers, managers and investors, while also leaving direction for future research.

First, to policy makers, our results show that ESG scores interact with parameters used for the monitoring of financial strength. It is therefore crucial to understand this nexus more deeply and use appropriate measures of financial strength in order to address and improve the incorporation of ESG values inside P&C insurers. (Policy makers worldwide are increasing the issuance as well as the stringency of rules addressed to tackle ESG issues in the insurance industry. For example, regarding the topic of climate change, the European Insurance and Occupational Pensions Authority (EIPOA) highlights the prominent role of the insurance industry in making the society and the economy more

climate resilient (https://www.eiopa.europa.eu/publications/role-insurers-tackling-climate-change-challenges-and-opportunities_en (accessed on 19 April 2023)).

In particular, for non-life-underwriting risks, the EIOPA works on guidelines to incorporate some climate-related measures in the solvency capital requirements (https://www.eiopa.europa.eu/system/files/2022-12/discussion_paper_on_the_prudential_treatment_of_sustainability_risks.pdf (accessed on 19 April 2023)). The United States Securities and Exchange Commission (SEC) recently released its climate disclosure requirements proposal (https://www.sec.gov/news/press-release/2022-46 (accessed on 19 April 2023)). The proposal concerns the move of the existing voluntary disclosures of climate-related risks to mandatory requirements that potentially carry increased legal liability. The Geneva Association outlines the pivotal role of insurance in promoting social sustainability, providing recommendations to insurance companies in order to foster socio-economic resilience and sustainability (https://www.genevaassociation.org/sites/default/files/2022-11/social_sustainability_report.pdf (accessed on 15 February 2022)). The China Banking and Insurance Regulatory Commission (CBIRC) in 2022 issued the green finance guidelines for banking and insurance sectors. These guidelines encourage insurers to improve their own ESG performance and promote an all-round green transition of economic and social development (https://www.cbirc.gov.cn/cn/view/pages/index/index.html (accessed on 19 April 2023)). In fact, every insurance company has an investment portfolio reported to the insurance regulatory authority. In the current economic landscape, insurance firms are among the major investors, meaning that improving their own ESG profile would contribute substantially to the progress of the whole economy to ESG values. The European Insurance and Occupational Pensions Authority (EIOPA) points out that, as long-term investors with an overall balance sheet of around EUR 8 trillion, insurers in the European Economic Area (EEA) can play a significant role in placing our economies on a more sustainable track (https://www.eiopa.europa.eu/system/files/2023-02/Factsheet%20-%20Green%20investments%202023v5.pdf (accessed on 10 June 2023)). Enhancing ESG values inside insurers would ultimately benefit economic growth and financial stability, because insurers play a key role in the propagation of systemic risk [39].

Second, based on our analysis of stock market returns, we provide important insights to insurance managers and investors. As we find that more sustainable insurance portfolios yield higher abnormal returns, we conclude that sustainable insurers create considerable value for shareholders. These excess returns suggest that sustainable insurers would be likely to face lower equity capital costs compared to less sustainable companies. This finding is particularly relevant, because equity capital plays a key role in the insurance business. In fact, an insurer raises capital which permits it to write an insurance policy. With its own capital plus the funds from insurance premiums, the insurer must pay out claims from the insurance policies and the associated business expenses. Evidently, the issuance and the cost of equity capital also have a considerable influence on other corporate decisions, like asset management strategies, and types of insurance to offer [8].

Finally, to follow-up research we leave the task to develop sound theoretical bases that would support our empirical results. One potential avenue of research would be to draw a model that explains the insurer's cost of capital as a function of its ESG rating. This setup should deliver predictions about the consequences on leverage and financial stability. A potential caveat of this research, as well as of future contributions in this field, may regard the implementation of suitable measures for the value and the financial health of insurers. In our analysis, we employed few indicators that are widely used in the literature, although we should note that scholars often debate on the appropriateness of such measures, as none of them lacks a certain degree of measurement bias. Another interesting extension of this research could involve the testing of ESG effects on proxies for insurer capital cost by separately identifying equity and debt costs. This approach would allow a more deep investigation of whether capital structures are insensitive to ESG, as our findings would suggest. Furthermore, it may also be worth to use the recent COVID-19 pandemic as an event study in order to test the hypothesis that ESG values provided an

effective competitive advantage to insurers that allowed them the survival against the recent turmoil. Unluckily, though, as most of insurance data are publicly available with annual frequency, it remains an evident issue to collect broad empirical dataset to address this hypothesis, which we leave to future work.

**Funding:** The APC was funded by the Free University of Bozen.

**Data Availability Statement:** Data were donwloaded from S&P Capital IQ and S&P Compustat.

**Conflicts of Interest:** The authors declare no conflict of interest.

## Appendix A

These are the insurance companies that we analyze:

Asia-Pacific: Anicom Holdings Inc., DB Insurance Co., Ltd., Dhipaya Group Holdings Public Company Limited, Dream Incubator Inc., Hyundai Marine & Fire Insurance Co., Ltd., ICICI Lombard General Insurance Company Limited, Insurance Australia Group Limited, MS&AD Insurance Group Holdings Inc., Meritz Financial Group Inc., QBE Insurance Group Limited, Samsung Fire & Marine Insurance Co., Ltd., Shinkong Insurance Co., Ltd., Sompo Holdings Inc., Suncorp Group Limited, The People's Insurance Company (Group) of China Limited, Tokio Marine Holdings Inc.

Europe: Admiral Group Plc, Alm. Brand A/S, Beazley Plc, Chubb Limited, Direct Line Insurance Group Plc, Linea Directa Aseguradora S.A., Sabre Insurance Group Plc, Tryg A/S.

Latin America and Caribbean: Qualitas Controladora S.A.B. de C.V.

Middle East: Qatar Insurance Company Q.S.P.C.

Unite States and Canada: AMERISAFE Inc., AXIS Capital Holdings Limited, Ambac Financial Group Inc., American Financial Group Inc., Arch Capital Group Ltd., Argo Group International Holdings Ltd., Assured Guaranty Ltd., Cincinnati Financial Corporation, Employers Holdings Inc., Erie Indemnity Company, Fairfax Financial Holdings Limited, HCI Group Inc., Hallmark Financial Services Inc., Heritage Insurance Holdings Inc., Hiscox Ltd, Intact Financial Corporation, James River Group Holdings Ltd., Kemper Corporation, Kinsale Capital Group Inc., Lancashire Holdings Limited, Loews Corporation, Markel Corporation, Mercury General Corporation, Old Republic International Corporation, Palomar Holdings, Inc., ProAssurance Corporation, RLI Corporation, Safety Insurance Group Inc., Selective Insurance Group Inc., The Allstate Corporation, The Hanover Insurance Group Inc., The Progressive Corporation, The Travelers Companies Inc., Trisura Group Ltd., United Fire Group Inc., United Insurance Holdings Corp., Universal Insurance Holdings Inc., W. R. Berkley Corporation, White Mountains Insurance Group Ltd.

**Table A1.** Definition of variables.

| Variable | Definition |
| --- | --- |
| *CR* | Combined ratio, i.e., the sum of incurred losses, loss adjustment expenses plus other underwriting expenses divided by earned premiums. |
| *E* | Natural logarithm of the company's environmental score. The environmental score is a discrete number in a range of 0–100. |
| *ESG* | Company's environmental, social, and governance (ESG) score. The ESG score is a discrete number in a range of 0–100. |
| *FL* | Financing leverage ratio, namely company's debt divided by equity capital. |
| *G* | Company's governance score. The governance score is a discrete number in a range of 0–100. |
| *HML* | High-minus-low [35]. |
| *MKTRF* | Excess market return [35]. |
| *MOM* | Momentum factor [36]. |

**Table A1.** *Cont.*

| Variable | Definition |
|---|---|
| *MTB* | Stock market price divided by book value per share. |
| *OL* | Operating (or underwriting) leverage ratio, namely net premiums written divided by the policyholders' surplus. Policyholders' surplus is company's assets minus liabilities. |
| *PRE* | Net premiums written divided by equity. |
| *r* | Monthly stock return |
| *S* | Company's social score. The social score is a discrete number in a range of 0–100. |
| *SIZE* | Natural logarithm of the company's total assets. |
| *SMB* | Small-minus-big [35]. |
| *ZSCORE* | Z-score, i.e., the sum of equity to total assets and return on average assets divided by the three-year standard deviation of the average return on assets. |

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
