# Peer review of "Effects from ESG Scores on P&C Insurance Companies"

_sustainability, doi:10.3390/su151612644_

Round 1
Reviewer 1 Report
After carefully reviewing your paper, my view is that it is an interesting paper with a potential to contribute to the literature. It can, however, be improved further as follows:
1. Introduction: Please clarify your research questions, objectives, background motivation, theoretical and empirical motivation and the lines of contributions to the literature. You can do this by sharply articulating your research questions/objectives, identify the potential theoretical, background and theoretical motivation or gaps, and explain how your study contributes to the literature.
2. Theoretical framework - An overarching theoretical framework is missing. You should provide a theoretical framework that supports the relationship between independent and dependent variables In doing so, please explicitly outline how it helps link the dependent and independent variables together by drawing on both seminal (old) and recently (newly) published studies.
3. Literature review: the literature review is very limited, I suggest to enrich it with the following studies that are directly related to this study
Gavana, G., Gottardo, P., & Moisello, A. M. (2022). Related party transactions and earnings management: The moderating effect of ESG performance. Sustainability, 14(10), 5823.
Velte, P. (2019). The bidirectional relationship between ESG performance and earnings management–empirical evidence from Germany. Journal of Global Responsibility, 10(4), 322-338.
Bătae, O. M., Dragomir, V. D., & Feleagă, L. (2020). Environmental, social, governance (ESG), and financial performance of European banks. Accounting and Management Information Systems, 19(3), 480-501.
Landi, G., & Sciarelli, M. (2018). Towards a more ethical market: the impact of ESG rating on corporate financial performance. Social responsibility journal, 15(1), 11-27.
Zhou, G., Liu, L., & Luo, S. (2022). Sustainable development, ESG performance and company market value: Mediating effect of financial performance. Business Strategy and the Environment, 31(7), 3371-3387.
4. Research design – Please identify, classify and explain your variables – dependent, independent and control variables, as well as any others, such as moderating or mediating variables. Please also explain your sample selection clearly (insert a table tabulating the steps - how many was missing, many had data, how many selected and why) and also clarify in a normative way how the variables are operationalised. Similarly, explain your sample in a tabular form, outlining step by step the total population to the selection of the final sample.
5. Conclusion – Please outline a summary of findings, contributions, implications, limitations and avenues for future research. Especially, expand the discussions relating to contributions, limitations and avenues for future research.
I hope the author will positively embrace these constructive suggestions as a way of taking this research forward.
Reviewer 2 Report
The paper analyzes the relationship between ESG scores of P&C insurers worldwide considering a large time span (2012-2022) and several dimensions of the insurers performance and financial conditions.
The specificity of the insurane business probably requires an additional paragraph in the literature. The way an insurance company (expecially P&C) finances its business (mainly through the collection of premiums) might significantly affect the interpretation of Financial Leverage.
It is not clear the composition of the sample (size, geographical distribution - expecially as in a specific session an analysis is dedicated to the US subsample). It is quite relevant as the insurance sector is strictly regulated and the regulations are quite different in different geographical area. At least a table and a paragraph of comments should be addet to describe the composition of the sample, together with an information about the size of it. Are observation provided for each of the year of the time span considered (from 2012 to 2022)? Please clarify.
I suggest to add the regression equation anyway.
Additionally, if the data are taken from the companies financial statements, also differences in the Financial Reporting rules might be commented because they could potentially affect the results.
Finally, the evolution of the regulation in certain area (for example in EU, the introduction of the NFD Directive/2014 probably had some booster effect on the ESG performance - and disclosure - also of insurers, as well as for all the other sectors, but affecting differently different geaographical area).
Finally, given the lenght of the time span, additional comments (or even control variables) should be included about the big phenomenon which impacted the insurance sector (both economics trends and the COVID).
Overall, I suggest to the author to expand and work a little bit on the discussion, with more comments of the results with a focus on the managerial implications and a business perspective (going more "beyond the statistical analysis and results").
Minor comment - In Table 1: Descriptive statistics - there is a variable (DE) which is not described or presented elsewhere. At the same time, the FL is missing, therefore I assume it is a mistake and the variable DE actually is the FL. Please make it consistent and uniform.
Very good English with only very few minor typos or sentencing issues (for example in the first sentence of part 4. Results).
Reviewer 3 Report
1. First, the authors simply use Table 10 to define the dependent variable and all the independent variables without any further explanation, which makes the manuscript difficult to read, since the definitions of the variables are not clear.
2. Second, the robustness test issue obviously comes into play when the authors try to estimate the effect of ESG of independent variable.
3. Based on the findings of statistical research, more policy implications are needed in the conclusions section.
This manuscript has some grammar mistakes, so please recheck this manuscript thoroughly.
Round 2
Reviewer 1 Report
The paper has been dramatically improved. Congrats